# Active Surveillance in Intermediate-Risk Prostate Cancer: A Review of the Current Data

**DOI:** 10.3390/cancers14174161

**Published:** 2022-08-27

**Authors:** Leandro Blas, Masaki Shiota, Masatoshi Eto

**Affiliations:** Department of Urology, Graduate School of Medical Sciences, Kyushu University, Fukuoka 812-8582, Japan

**Keywords:** active surveillance, conservative management, intermediate-risk, prostate cancer

## Abstract

**Simple Summary:**

AS is an option for the initial management of selected patients with intermediate-risk PC. The proper way to predict which men will have an aggressive clinical course or indolent PC who would benefit from AS has not been unveiled. Genetics and MRI can help in the decision-making, but it remains unclear which men would benefit from which tests. In addition, there are several differences between AS protocols in inclusion criteria, monitoring follow-up, and triggers for active treatment. Large series and a few RCTs are under investigation, and more research is needed to establish an optimal therapeutic strategy for patients with intermediate-risk PC. This study summarizes the current data on patients with intermediate-risk PC under AS, recent findings, and discusses future directions.

**Abstract:**

Active surveillance (AS) is a monitoring strategy to avoid or defer curative treatment, minimizing the side effects of radiotherapy and prostatectomy without compromising survival. AS in intermediate-risk prostate cancer (PC) has increasingly become used. There is heterogeneity in intermediate-risk PC patients. Some of them have an aggressive clinical course and require active treatment, while others have indolent disease and may benefit from AS. However, intermediate-risk patients have an increased risk of metastasis, and the proper way to select the best candidates for AS is unknown. In addition, there are several differences between AS protocols in inclusion criteria, monitoring follow-up, and triggers for active treatment. A few large series and randomized trials are under investigation. Therefore, more research is needed to establish an optimal therapeutic strategy for patients with intermediate-risk disease. This study summarizes the current data on patients with intermediate-risk PC under AS, recent findings, and discusses future directions.

## 1. Introduction

Worldwide, prostate cancer (PC) is the second most commonly diagnosed cancer after lung cancer and the fifth cause of death by cancer in men [1]. In 2020, there were an estimated 1.4 million new cases diagnosed with PC and 375,000 deaths worldwide. With the growth of the aging population, the number of PC cases is expected to increase by 3.5 times by 2040 worldwide [2].

Active surveillance (AS) is a monitoring strategy to avoid or defer curative treatment, minimizing treatment-related toxicity without compromising survival. AS consists of long-term follow-up with evaluation of prostate-specific antigen (PSA), imaging, and prostate biopsy. AS allows appropriate risk reclassification and patient selection for intervention. In the last decades, AS has become a standard of care for men with low-risk PC (≤cT2a, Gleason score [GS] ≤ 6, and PSA < 10 ng/mL) [3,4,5]. Moreover, a trend toward the increased use of AS has been observed in patients with low and intermediate-risk PC. For example, in the United States (US), low- and intermediate-risk patients choosing AS increased from 14.5% in 2010 to 42.1% in 2015, and from 5.8% to 9.6%, respectively [6]. In a study from Sweden that included 98% of newly diagnosed PC from 2009 to 2014, the AS use increased from 57% to 91% for very low-risk PC, from 40% to 74% for low-risk, but remained at approximately 19% for intermediate-risk PC [7].

Some patients with intermediate-risk PC have an aggressive clinical course and require active treatment, while others have indolent disease and may benefit from AS. However, the proper way to differentiate between both groups of patients has not been unveiled. This study summarizes the current data on patients with intermediate-risk PC under AS, recent findings, and discusses future directions.

## 2. Evidence on Non-Active Treatment in Intermediate-Risk PC

### 2.1. Prognosis in Intermediate-Risk PC by Observation vs. Active Treatment

To date, the Prostate Cancer Intervention versus Observation Trial (PIVOT [8]) and the Scandinavian Prostate Cancer Group 4 Study (SPCG-4 [9]) have investigated immediate treatment (radical prostatectomy [RP] or radiation therapy [RT]) versus observation (AS or watchful waiting) in localized PC. In addition, the three-arm Prostate Testing for Cancer and Treatment (ProtecT [10]) compared RP versus RT versus observation in this setting. Additionally, other non-randomized studies compared AS versus intervention [11,12]. Table 1 shows a summary of the studies.

In men with intermediate-risk of the PIVOT, the 10-year overall survival (OS) was higher (71% vs. 62%) undergoing RP than observation. Similarly, in men with intermediate-risk PC in the SPCG-4 study, RP was associated with an absolute reduction in overall mortality (15.5 percentage points), PC death rate (24.2 percentage points), and risk of metastases (19.9 percentage points) [9]. Meanwhile, in the ProtecT study, patients in observation presented a higher rate of metastases than RP and RT, probably due to intermediate and high-risk PC included in the observation group. However, there was no significant difference in 10-year survival outcomes. Thus, previous studies showed inferior results in intermediate-risk PC when non-active treatment was chosen.

However, some limitations of these three randomized controlled trials (RCT) include watchful waiting or monitoring less close than current AS, without confirmatory and serial biopsies. In addition, there was no opportunity for active treatment or use of magnetic resonance imaging (MRI). Additionally, the PIVOT trial was initiated before the PSA era. Thus, the findings from these studies cannot be applied to the current AS strategy.

### 2.2. Oncological Outcomes by AS in Intermediate-Risk versus Low-Risk PC

Many series have compared the outcomes among intermediate- and low-risk patients [13]. Musunuru et al. demonstrated that 15-year metastasis-free survival (MFS), OS, cancer-specific survival (CSS), and treatment-free survival were inferior in the intermediate-risk group than the low-risk, with more than three times increased risk of metastasis at 15-year follow-up (hazard risk [HR] 3.14, 95% confidence interval [CI] 1.51, 6.53; *p* = 0.001) [14]. Similarly, a study from the Veterans Health Administration included 9733 men initially managed under AS (*n* = 1003, 10.3% with intermediate-risk disease, of whom 76.8% had favorable- and 23.2% unfavorable-risk disease) [15]. With a median follow-up of 7.6 years, the 10-year cumulative incidence of metastasis and PC-specific mortality were higher for patients with intermediate-risk (favorable- and unfavorable-risk) than in low-risk disease.

A systematic review and meta-analysis comparing results of 17 AS series in low- and intermediate-risk showed worse CSS in the intermediate-risk group after 10 years (odds ratio [OR] 0.47; 95% CI, 0.31–0.69) and 15 years (OR 0.34; 95% CI, 0.2–0.58) [16]. There was no statistical difference in 5-year OS (OR 0.84; 95% CI, 0.45–1.57), but 10-year OS was worse in the intermediate-risk group (OR 0.43; 95% CI, 0.35–0.53). Similarly, there was no statistical difference in 5-year MFS (OR 0.55; 95% CI, 0.2–1.53), but MFS was worse in the intermediate-risk group after 10 years (OR 0.46; 95% CI, 0.28–0.77). Thus, this evidence suggests that AS outcomes for intermediate-risk and low-risk are comparable in short-term and medium-term follow-up, but poorer in the long term.

### 2.3. Patients-Reported Outcomes in AS

Living with untreated PC causes anxiety and uncertainty. Men with intermediate-risk PC likely have a higher anxiety rate than low-risk patients. A study showed that 29% of men in AS presented mild PC-specific anxiety during the first year, and the rate decreased significantly with time [17]. However, only 22 of 413 (5%) patients had GS 7, and a separate analysis was not performed. A systematic review including 34 studies suggested no differences in anxiety and depression rates for up to five years between men on AS and active treatment (RP or RT) [18].

A systematic review of 13 qualitative studies on factors that influence the decision-making between AS and active treatment showed that the decision of AS was an ongoing behavior (not a punctual choice) and included their assessment of risk, the influence of family and friends, beliefs about treatment, and doctor and system factors [19]. A scoping review in men undergoing AS identified interventions that affect the psychosocial burden, such as lifestyle, education, information, coping, and psychosocial support [20]. Interventions that appear to decrease psychosocial burden include psychosocial support involving the family and spouse in the decision-making, education, and tailored information on treatment options [20]. In addition, they recommend the assessment and promotion of effective coping and self-management strategies.

## 3. Risk Stratification of Intermediate-Risk PC in AS

A systematic review including men with intermediate-risk PC found high variability in outcomes, with adverse surgical pathology ranging from 15 to 64% and 5-year disease progression of 21–91%, showing that outcomes in intermediate-risk PC are heterogeneous [21]. The distinction between favorable and unfavorable intermediate-risk is based on a study that analyzed 1024 men with National Comprehensive Cancer Network (NCCN) intermediate-risk PC who were treated with radiation therapy (RT) alone or in addition to androgen deprivation therapy [22]. Primary Gleason pattern 4, ≥50% rate of positive cores number, and two or more than among clinical-stage T2b–2c, PSA 10–20 ng/mL, and GS 7, were predictors of increased distant metastasis [22]. Thus, NCCN guidelines categorize intermediate-risk as favorable if present all the followings: <50% biopsy cores positive, GS ≤ 3+4, only one between (PSA 10–20 ng/mL, cT2b–cT2c, and GS 3+4).

Favorable intermediate-risk PC is also a heterogeneous group. For example, a patient with one positive core of GS 6 and a PSA of 12 ng/mL (due to an enlarged prostate) and a patient with two cores of GS 3+4 with 40% of Gleason pattern 4 and a PSA below 10 ng/mL, are both categorized as favorable intermediate-risk PC. However, both have different risks of adverse pathological findings at surgery. Studies have shown no difference in 15-year MFS between men with GS 6 and PSA between 10 and 20 ng/mL and men with low-risk PC (GS 6 and PSA < 10 ng/mL) [14]. On the other hand, men categorized as intermediate-risk men due to a GS 3+4 have a higher risk of adverse pathology [14]. Another study compared adverse pathology (upgrading or upstaging to ≥pT3a) on RP specimens in intermediate-risk PC men under AS. A total of 382 of 1731 (22.1%) men who had intermediate-risk due to PSA 10–20 ng/mL (and a GS 6) and 2340 of 8367 (28%) men who were categorized as intermediate-risk due to GS 3+4 (and a PSA < 10 ng/mL) presented adverse pathology at RP [23]. On the multivariable analysis, a PSA level of 10–20 ng/mL had lower odds of harboring an adverse pathology (versus PSA < 10 ng/mL, OR 1.87, 95% CI 1.71–2.05, *p* < 0.001) than GS 3+4 men (versus GS 6, OR 2.56, 95% CI 2.40–2.73; *p* < 0.001) [23]. Thus, the presence of Gleason pattern 4 increased the metastatic rate, but not a PSA level between 10 and 20 ng/mL [13].

Therefore, risk stratification is essential to implement AS for patients with intermediated-risk PC by utilizing the risk factors described below.

### 3.1. High-Volume GS 6

A study including 6775 (*n* = 1288 with intermediate-risk) men on AS evaluated the factors related to conversion to active treatment [24]. With a median follow-up of 6.7 years, 2260 (33.4%) patients converted to active treatment. Interestingly, conversion rates for high-volume GS 6 (defined as ≥4 biopsy cores involved) men were higher than the intermediate-risk disease (63.6% versus 38.3%, with a 5-year conversion-free probability of 35.8% and 64.1%, respectively) and similar to other high-risk men. In addition, a study including 561 men (25% intermediate-risk) under AS showed that an increasing percentage of positive core involvement was an independent predictor of progression (HR, 1.03) [25].

### 3.2. Percentage of Gleason 4 Pattern

Another main factor is the percentage of Gleason 4 pattern. A study showed that men with GS 3+4 with less than 5% of pattern 4 on prostate biopsy presented similar GS, pathologic stages, total tumor volume, and insignificant tumor rate at RP to those men who had GS 3+3 [26]. In addition, patients with GS 3+4 with less than 5% of pattern 4 presented a high rate of downgrade at the RP specimen. Additionally, several studies have shown that the percentage of pattern 4 in biopsy samples is a predictor of pathological T3 on RP specimens and biochemical recurrence disease [27]. On the other hand, a study that evaluated 608 men with low-volume intermediate-risk PC (1 or 2 cores of GS 3+4 and PSA < 20 ng/mL) undergoing RP showed that approximately 25% presented GS ≥ 4+3, seminal vesicle invasion, or lymph-node involvement [28]. Moreover, they could not identify any presurgical clinical or pathological criteria that could identify a subgroup of the low-volume intermediate-risk PC with similar rates of adverse pathologic findings with those of low- and very-low risk cohorts. Based on these findings, Klotz et al. suggested that men with GS 3+4 with <10–20% of Gleason pattern 4 may be considered for AS while patients with GS 3+4 with >20% Gleason pattern 4 or GS 4+3 disease should be treated [13].

Moreover, patients with PC with intraductal carcinoma or cribriform pattern histology have more aggressive behavior and an increased risk of metastasis and PC-specific mortality [29,30]. European Association of Urology (EAU) guidelines discourage AS for patients with these findings [3].

### 3.3. PSA Density

Some protocols include PSA density (the ratio between PSA and prostate size). A systematic review and meta-analysis performed on low-risk men under AS showed that higher PSA density was associated with an increased risk of upgrade [31]. In addition, a study in intermediate-risk men showed that an increased PSA density predicted adverse pathologic findings at RP [28]. While a high PSA density is not specific enough to exclude a patient from AS, men with a higher PSA density should be evaluated with MRI, targeted, and systematic biopsies [13].

### 3.4. Race/Ethnicity

African Americans with low-risk PC on AS have a higher risk of grade or volume progression than Caucasian Americans [32,33,34,35]. In a study from the Surveillance, Epidemiology, and End Results (SEER) Prostate AS/Watchful Waiting database, only black men with GS 6 presented increased cancer-specific mortality compared to nonblack men [36]. Treatment disparities and access to health care may play an important role in these clinical racial disparities [37]. On the other hand, other studies have shown similar rates of upstaging or upgrading [38]. A recent study from the SEARCH database involving 355 African American and 540 Caucasian men with low-risk treated with RP showed no significant difference in upgrading, upstaging, or biochemical recurrence [39]. Thus, there is controversial evidence on upstaging or upgrading in African Americans. Therefore, guidelines recommend AS in African America men, advising of a possible higher risk of significant cancer.

### 3.5. Age

Age is associated with an increased risk of higher grade and a higher prevalence of PC [13]. A study including 1433 on AS (17.2% with intermediate- or high-risk disease) showed that younger age at diagnosis was independently associated with a lower risk of GS upgrade on biopsy during AS but not with the risk of active treatment or biochemical recurrence after RP in the intermediate term [40]. The benefit of avoiding active treatment in young men opting for AS are more substantial (preserving sexual function and continency) than in older men. However, as long-term outcomes in intermediate-risk disease men are worse than in low-risk men, AS for young men should be offered with caution and close, and long-term monitoring is required.

### 3.6. Genetic Tests

The US Food and Drug Administration has approved four tissue biomarkers Decipher^®^ (GenomeDX, Vancouver, BC, Canada), Oncotype DX^®^ (Genomic Health, Redwood City, CA, USA) Prolaris^®^ (Myriad Genetics Inc., Salt Lake City, UT, USA) and ProMark score^®^ (Meta Mark Genetics, Waltham, MA, USA), and are commercially available in some countries. Decipher^®^ (GC; GenomeDX, Vancouver, BC, Canada) genomic classifier score (GCS) uses a transcriptome microarray assay. Twenty-two RNA biomarkers have been associated with more aggressive PC and metastasis prediction after RP. The GCS was associated with short-term biopsy GS upgrading among patients on AS (OR 1.37 per 0.1-unit increase, 95% CI 1.05–1.79; *p* = 0.02) [41]. In addition, GCS was an independent predictor of adverse pathology (OR 1.29 per 10% increase, 95% CI 1.03–1.61; *p* = 0.025, and 1.34 per 0.1-unit increased; *p* = 0.002) [42,43]. A high GCS (>0.6) increases the risk of PC-specific mortality (OR adjusted for Cancer of the Prostate Risk Assessment post-Surgical score of 3.91, 95% CI 2.43–6.29) within 10 years after RP [44]. A meta-analysis including five studies in men with PC post RP showed that GCS was a predictor for metastasis in the multivariate analysis (HR, 1.30 per 0.1-unit increase; 95% CI, 1.14–1.47; *p* < 0.001) [45].

Oncotype DX^®^ (GPS; Genomic Health, Redwood City, CA, USA) test using reverse transcriptase-polymerase chain reaction (RT-PCR) assesses 12 PC-related genes and five housekeeper genes. The resulting Genomic Prostate Score (GPS, 0 to 100) can be used on prostate specimens or biopsy tissue with more than 1 mm PC. A higher initial GPS indicates an increased risk of upgrading and adverse pathology [46]. Some studies that associated GPS and MRI Prostate Imaging-Reporting and Data System (PIRADS) with systematic and targeted biopsies [47,48,49] reported that the GPS was predictive of adverse pathology on the RP specimen, independently of MRI scores. In addition, GPS widely varied among men with GS 6, and GPS did not correlate with PIRADS version 1, but there was a wide and overlapping PIRADS version 2 among men with low- to intermediate-risk PC [48]. In addition, GPS usage increased the enrollment of patients on AS [50]. However, a recent multicenter prospective-retrospective from the Canary PASS that included 432 patients under AS found no association between the GPS and adverse pathology at RP or upgrade on the following biopsy [51].

Prolaris^®^ (Myriad Genetics Inc., Salt Lake City, UT, USA) uses RT-PCR to assess 31 cell cycle progression pathways and 15 housekeeper genes. The gene expression level is included in an algorithm to calculate the score, the majority between 1 and 11. A higher score is associated with a poor prognosis. This test has been validated as a prognostic factor after RP, but no studies were conducted on AS cohort.

ProMark score^®^ (Meta Mark Genetics, Waltham, MA, USA), uses quantitative multiplex proteomics in situ imaging system to identify and measure eight protein-based biomarkers. A higher score (from 0 to 100) is related to a more aggressive and lethal PC. An advantage is the lack of need for a pathologist, given that can objectively detect high-grade molecular features in small tissue samples. This score added prognostic value relative to NCCN and D’Amico risk stratification systems [52]. However, no supplementary study has been conducted.

An observational study including 747 tested by Decipher Prostate Biopsy (*n* = 227), Oncotype DX Prostate (*n* = 81), and Prolaris (*n* = 439) have shown that genetic tests have changed the management of patients with favorable-risk cancer in AS with a number needed to test of 8.8 and 25.9 to shift one patient to AS and primary therapy, respectively [53].

NCCN guidelines recommend considering Decipher, Oncotype DX Prostate, or Prolaris, during initial risk stratification for patients with low- or favorable intermediate-risk and life expectancy >10 years who are candidates for AS or definitive therapy [4]. The American Society of Clinical Oncology (ASCO)-EAU-American Urological Association (AUA) recommends its use only when the outcome is likely to affect the management [54]. In multivariate analysis, genetic testing has significantly improved prognostic accuracy in differentiating patients with indolent disease and clinically significant PC [3,54]. However, it is not clear which test is adequate to benefit which patient. There is a need for more prospective validation of these markers in men undergoing AS for low to intermediate-risk diseases.

### 3.7. Genetic Alterations

Some genetic mutations are associated with more aggressive diseases. Patients with germline mutations in tumor-suppressor genes such as *BCRA1*, *BRCA2,* and *ATM* gene, that are associated with homologous recombination repair, are at higher risk of progression and grade reclassification in patients undergoing AS [55]. In a study that included 1211 men under AS (*n* = 26 with GS 7), 11 of 26 carriers of *ATM*, *BRCA1,* or *BRCA2* experienced grade reclassifications, and 278 of 1185 non-carriers (adjusted HR 1.96, 95% CI 1.004–3.84; *p* = 0.04). Moreover, reclassification was higher in men with *BRCA2* gene mutation with an adjusted HR of 2.74 (95% CI 1.26–5.96; *p* = 0.01). In addition, *HOXB13* gene mutation [56], mutations in other genes for DNA repair such as *CHEK2* or *MSH2*, and single-nucleotide polymorphisms predispose toward more aggressive cancers [16,57]. A study that evaluated tissue from the initial biopsies of low-risk disease men showed *PTEN* loss in 29 of 190 men (15%) on PRIAS [58]. *PTEN* loss was significantly associated with upgrading on confirmatory biopsy (HR, 2.57; 95% CI, 1.16–5.70; *p* = 0.02), change to active treatment (HR, 2.31; 95% CI, 1.26–4.19; *p* = 0.006), and adverse RP findings (HR, 4.75; 95% CI, 1.84–12.23; *p* = 0.001). However, current AS programs do not include genetic tests or these risk factors in the decision-making. Further investigations on the role of genetic alterations in AS are needed.

### 3.8. New Imaging Modalities

MRI has improved the detection of clinically significant PC in men on AS [3]. Recently, several cohorts have included regular MRIs in AS protocols. It is used at the start to confirm AS eligibility and recommended before any subsequent biopsies, and some studies have introduced it earlier in the diagnostic pathway [59,60]. NCCN guidelines recommend for men with intermediate-risk disease considering AS, if MRI was not performed initially, consider MRI with or without prostate biopsy and/or molecular analysis. The Cancer Care Ontario (CCO)-ASCO guidelines recommend offering MRI or genetic tests when clinical and pathological findings are discordant and could be useful in identifying occult cancers or changes indicative of progression. Moreover, recommend offering these tests when the decision between AS and active treatment is uncertain such as in low-volume GS 3+4 [5].

Prostate-specific membrane antigen (PSMA) positron emission tomography (PET) has proven its diagnostic value in localized PC, with high sensitivity, specificity, and negative predictive values that suggests a potential role in the evaluation of men on AS. However, studies on patients under AS have not been conducted [61]. In addition, micro ultrasound-guided biopsy and MRI/PET have also shown promising results in detecting significant PC, but larger studies are needed [61,62].

### 3.9. Guideline Recommendations Based on Risk Factors

Accordingly, the CCO-ASCO guideline encourages AS in highly selected patients with low-volume, intermediate-risk GS 3+4 localized PC [5]. The EAU guideline offers AS to men with <10% Gleason pattern 4, accepting the potential increased risk of further metastasis [3]. The NCCN guideline state that favorable intermediate-risk men with a life expectancy >10 years may be considered for AS, particularly in men with a low percentage of Gleason 4 pattern, low tumor volume, low PSA density, and/or low genomic risk (from tissue-based tumor molecular analysis) [4].

## 4. Criteria for Inclusion, Monitoring, and Trigger for Intervention

### 4.1. Inclusion Criteria for AS in Intermediate-Risk PC

Many cohorts have included men with intermediate-risk under AS [14,24,25,63,64,65,66,67,68,69,70,71,72,73,74,75,76,77,78,79]. Table 2 shows a summary of studies including intermediate-risk patients undergoing AS and compares oncological outcomes between low-risk and intermediate-risk. Among intermediate-risk men, there are differences regarding patient selection criteria, monitoring, and trigger for active treatment. Inclusion criteria varied between studies. Most series used the D’Amico intermediate-risk criteria: GS 7 and/or PSA 10-20 ng/mL, and cT1–2. Some series used the EAU criteria of intermediate-risk PC: GS ≤ 7, PSA < 20 ng/mL, and ≤cT2b. Other studies used the Cancer of the Prostate Risk Assessment (CAPRA) score intermediate-risk criteria, and in some articles, the differentiation was limited to GS 3+4 disease.

A systematic review including 264,852 patients under AS showed that a minority of AS protocols included MRI for recruitment, follow-up, or reclassification [80]. More than half of the protocols included men with intermediate- or high-risk PC, approximately half excluded low-risk patients with >3 positive cores, and about 40% excluded men with core involvement >50% per core.

Selection criteria for AS are limited due to the lack of RCTs. Further research to unify these criteria is needed.

### 4.2. Monitoring and Triggering in Intermediate-Risk PC

Current guidelines about monitoring during AS do not distinguish between low- and intermediate-risk men. Thus, the monitoring protocol is the same for a patient with one core of GS 3+3 as for a man with GS 3+4 with a PSA level of 15 ng/mL. Include regular PSA, digital rectal examination, and confirmatory biopsy within 18 months (Table 1). A systematic review showed that 80% of AS protocols mandated a confirmatory biopsy and only 10.3% mandated triggered biopsies [80].

A study that evaluated the rate of loss of follow-up in 2211 men under AS in 44 centers found that with a median surveillance period of 32 months, the estimated 2-year loss of follow-up was 10% [81]. A higher Charlson comorbidity index (HR 1.55, 95% CI 1.08–2.23) and African American men (HR 2.77, 95% CI 1.81–4.24) were independent risk factors for loss of follow-up. Moreover, the rate of loss of follow-up was 9.2% in men with GS 6, but 11.3% in patients with GS 7. On the other hand, a systematic review and meta-analysis showed a comparable proportion of patients who remained on AS between the low- and intermediate-risk groups after 10 and 15 years of follow-up (OR 0.97, 95% CI, 0.83–1.14; and OR, 0.86; 95% CI, 0.65–1.13, respectively) [16]. This suggests that intermediate-risk PC have similar or higher rates of loss of follow-up than low-risk. However, as intermediate-risk men in AS have an increased risk of metastasis and PC death, a risk-adapted strategy for follow-up testing is needed to increase AS adherence, especially in the long term.

Studies have shown that MRI is not an effective substitute for systematic biopsy and should not be used as a replacement for confirmatory biopsy [5]. A systematic review and meta-analysis showed MRI would have missed PC upgrading in 10% to 15% upgrades [82]. A cohort study showed that using only an increased PIRADS score or clinical stage to trigger a biopsy result in avoiding 681 unnecessary biopsies per 1000 men but missing GS 3+4 or higher disease in 169 of 1000 men [83]. Another systematic review including 45 observational studies (some of them including intermediate-risk men) showed that most studies did not clearly define or report adherence to monitoring protocols [84].

Criteria used for switching to active treatment in intermediate-risk disease included histological evidence of progressive disease (upgrade to GS 4+3 or higher, invasive cribriform and intraductal carcinoma, >50% of biopsies positive cores [multiple positive cores from the same lesion on MRI count for one positive core]) or clinical progression [16].

## 5. Intervention during AS

### 5.1. Non-Medical Intervention

Anxiety is one of the most common effects of AS, and most men under AS want to have an active role instead of only waiting for the tests. Small, randomized trials compared interventional behavior versus controls in patients with localized PC under AS. The Prostate Cancer Lifestyle Trial randomized patients with low-risk PC to a low-fat plant-based diet, physical activity, and stress management, and to attend group support sessions for 12 months (*n* = 44) versus no lifestyle intervention (*n* = 49) [85,86]. After two years of follow-up, patients with the intervention were associated with a reduced risk of progression to treatment (27% control and 5% experimental had undergone conventional PC treatment; *p* < 0.05), and decreased PSA and cholesterol levels.

The ERASE trial randomized men with very-low to intermediate-risk PC on AS to a (12 weeks of thrice-weekly) high-intensity interval training (*n* = 26) or usual care group (*n* = 26) during AS [87]. Patients in the intervention arm showed decreased PSA level (−1.1 μg/L, 95% CI, −2.1 to 0.0; *p* = 0.04), and PSA velocity (−1.3 μg /L/y, 95% CI, −2.5 to −0.1; *p* = 0.04) compared with controls. No statistically significant differences were found in PSA doubling time or testosterone level.

On the other hand, the ALLIANCE study performed in 91 centers in the US randomized men in AS to 1:1 to a behavioral intervention (*n* = 226) that promoted daily consumption of targeted seven or more vegetable-fruit servings or control group (*n* = 217) [88]. At 24 months of follow-up, there was no difference in patient-reported scores, including anxiety (general prostate and anxiety related to PSA levels, and fear of recurrence). These small studies suggest that a low-fat plant-based diet, combined with physical exercise and stress management reduces the chances of progression. Larger studies and longer follow-ups are needed to prove the effect of lifestyle change in patients with intermediate-risk PC under AS.

### 5.2. Medical Intervention

Recently, the ENACT trial, a phase 2, open-label, randomized 1:1 enzalutamide as monotherapy (*n* = 114, intermediate-risk *n* = 53) 160 mg for 12 months plus AS or continued AS alone (*n* = 113, intermediate-risk *n* = 53) in patients with localized low and intermediate-risk PC with less than prior six months on AS [89]. With a follow-up of two years after treatment, enzalutamide presented a reduced risk of progression with an HR of 0.54 (95% CI, 0.33–0.89; *p* = 0.02). In addition, enzalutamide increased 3.5 times the odds of a negative biopsy and significantly reduced the percentage of cancer-positive cores. Moreover, enzalutamide delayed PSA progression by six months with an HR of 0.71 (95% CI, 0.53–0.97; *p* = 0.03). Ninety-two percent of men in the enzalutamide arm presented grade ≤ 3 adverse events, most commonly fatigue (55.4%) and gynecomastia (36.6%).

The ProVent (NCT03686683) study is comparing the effect of Sipuleucel-T on histopathologic reclassification to higher GS or upstage in men within 12 months of AS (low and intermediate-risk) versus standard AS. In addition, a prospective cohort study (NCT04549688) is investigating the effect of high-intensity focused ultrasound on post-treatment systematic and targeted biopsies.

Two phase 2 trials (NCT NCT04597359 and NCT04300855) are investigating the effect of green tea catechins versus placebo on progression. In addition, a study (NCT03679260) is evaluating a carbohydrate restriction diet intervention on mean difference change in proliferative index (measured by Ki67/apoptosis rate), and (NCT02095145) is investigating the effect of pomegranate extract pill on tumor growth in patients under AS.

In addition, the Movember Foundation’s Global Action Plan Prostate Cancer AS (GAP3) cohort comprises the largest global, centralized database with more than 21,500 patients from 27 cohorts [90] aims to develop worldwide clinical guidelines for AS. This international project is also analyzing additional parameters such as imaging modalities, biomarkers, and genomics. Similarly, the Prostate Cancer Active Surveillance Project (PCASP), the largest group examination of utilization and quality of AS for intermediate-risk PC, aims to develop a new gold standard of AS guidelines and practices. Moreover, the Prostate Cancer Active Surveillance Trigger trial (PCASTt/SPCG-17), a trial including centers in Sweden, Norway, Finland, and the UK, will randomize men in AS to receive current practice or standardized triggers based on PSA density and MRI [91]. These studies will address many of the questions on how to optimize current AS protocols.

This article has potential limitations, such as a high risk of bias, mainly due to its subjectivity and lack of methodology.

## 6. Conclusions

In conclusion, survival outcomes for intermediate-risk men on AS are comparable to low-risk in short-term and medium-term follow-up, but poorer in the long term. Therefore, the optimization of AS protocols on inclusion criteria, monitoring, and triggering active treatment for intermediate-risk PC is required by risk stratification using clinical parameters, genetic, and imaging tests. Furthermore, the development of an intervention to reduce the risk of disease progression during AS is important, especially for intermediate-risk PC.

## Figures and Tables

**Table 1 cancers-14-04161-t001:** Active surveillance versus other active treatment in localized PC.

Authors	Study Name	Number of Patients Intermediate Risk/Total, *n* (%)	Type	Initiation	Comparator	Gleason 4	Median Follow-Up	PC Mortality	Non-PC Mortality	Reference Number
Hamdy et al.	ProtecT	490/1634 (31%)	Prospective RCT	2001–2009	AS vs. PR vs. RT	NA	10 years	Similar deaths per 1000 person year of 1.5, 0.9 and 0.7 for AS, RP, and RT, respectively	Similar all cause mortality per 1000 person year AS = 10.9; RP = 10.1; and RT = 10.3	[10]
Wilt et al.	PIVOT	Observation = 120/348 (34.5%) RP = 129/383 (33.6%)	Prospective RCT	1994–2002	RP vs. observation (WW)	NA	12.7 years	Slightly higher 10-year PC mortality in RP (9.0% vs. 8.6%)	Higher 10 year mortality in AS (71.2% vs. 62.6%)	[8]
Bill-Axelson et al.	The Scandinavian Prostate Cancer Group 4 Study	Observation = 133/348 (38.2%) RP = 148/347 (42.7%)	Prospective RCT	1989–1999	RP vs. observation (WW)	54/116 (46.5%)	13.4 years	Higher number of deaths by PC during follow-up in WW (99 vs. 63)	Higher number of deaths by any cause during follow-up in WW (247 vs. 200)	[9]
Thomsen et al.	Active surveillance versus radical prostatectomy in favorable-risk localized prostate cancer	AS = 271/647 (42%) RP = 276/647 (43%)	Retrospective	2002–2012 for AS 1995–2011 for RP	RP vs. AS	NA	8.6 years	Slightly higher 10-year PC mortality in RP (1.5% vs. 0.4%)	Slightly higher 10-year non-PC mortality in RP (12.0% vs. 10.7%)	[12]
Stattin et al.	Outcomes in localized PC: National PC Register of Sweden follow-up study	AS/WW = 936/2021 (42%) RP = 2172/3399 (52.5%)	Retrospective	1997–2002	RP vs. AS/WW	NA	8.2 years	Higher 10-year PC mortality in AS/WW (5.2% vs. 3.4%)	Higher 10-year non-PC mortality in AS (23.4% vs. 11.3%) *	[11]

AS, active surveillance; NA, not available; PC, prostate cancer; RCT, randomized control trial; RP, radical prostatectomy; RT, radiotherapy; WW, watchful waiting. * Include low- and intermediate-risk cohorts.

**Table 2 cancers-14-04161-t002:** Comparison of Active Surveillance outcomes between intermediate-risk and low-risk disease.

	Authors	Institution	Patients, n (%)	Type	Inclusion Criteria for IR	Gleason 4, n (%)	Follow-Up Protocol	Trigger for Intervention	Median Age in Years	Median Follow-Up	Continued on AS	Outcomes	Reference Number
IR	Musunuru et al.	University of Toronto	213/945 (22.5%)	Prospective	PSA 10–20 ng/mL, GS 3+4	128 (60%)	Confirmatory biopsy at 6–12 months and then every 3–4 years. PSA every 3 months for 2 years and then every 6 months	Upgrade. PSADT	72	6.7 years	61% at 10-year, 48% at 15 years	10 and 15-year CSS, 97% and 89%; 10, and 15- years OS, 67% and 51%; 10 and 15-year, MFS 91% and 82%	[14]
LR			732/945 (77.5%)						67	6.5 years	64% at 10-years, 58% at 15 years	10 and 15-year CSS, 98% and 97%; 10 and 15-year OS, 84% and 67%; 10 and 15-year, MFS 96% and 95%	
IR	Cooley et al.	Multi-institutional	1288/6775 (19.0%)	Retrospective	cT2, PSA 10–20 ng/mL, GS 3+4	563 (43.7)	Varied among institutions.	−	64	6.1 years	64.1% at 5- years	CSS, 99.8%; OS, 98.6% *	[24]
LR			4604/6775 (67.9%)							6.8 years	78.6% at 5- years		
IR	Savdie et al.	Vancouver Prostate Centre	144/651 (22.1%)	Retrospective	≤T2, PSA <20 ng/mL, GS 4+3	65 (45.1)	Confirmatory biopsy within 18 months, then every 1–2 years, DRE and PSA every 6 months.	Upgrade, upstage, PSA DT	67.2	4.4 years	50% at 5-year, 34.1% at 10-year	CSS, 99.3% for IR; OS, 97.7% (5- year and 10-year OS, 98.6% and 94.1%) *	[25]
LR			262/651 (40.2%)						64.4	4.5 years	55.5% at 5-years, 38.8% at 10-years	CSS, 99.6%	
IR	Bokhorst et al. **	PRIAS	31/5302 (1%)	Prospective	≤T2c, PSA <20 ng/mL, GS 3+4 without invasive CR/IDC, ≤50% of PPC	31 (0.5)	Confirmatory biopsy at 1, 4, 7, and 10 years then every 5 years. PSA every 3, and DRE every 6 months for 2 years, then PSA every 6 months and DRE once a year.	Upgrade, cribriform or intraductal carcinoma, >50% PPC, upstage ≥ cT3	65.9	10 years	48% at 5-years, 27% at 10 years *	10-year CSS > 99% *	[63]
IR	Masic et al.	University of California San Francisco	188/1243 (15.1%)	Prospective	≤T2, PSA <20 ng/mL, GS 3+4, CAPRA	124 (65.9)	Confirmatory biopsy at 1 year, then every 1–2 years, 2 PSA every 3 months. TRUS every 6 months.	Upgrade	62 *	62 months	49% at 5-years	CSS, 100%; 5-year MFS, 98%	[64]
LR			1011/1243 (84%)								64% at 5-years	CSS, 100%; 5-year MFS, 99%	
IR	Selvadurai et al. **	Royal Marsden	88/471 (18.6%)	Prospective	>65 y, ≤T2, PSA <15 ng/mL, GS 3+4, and PPC < 50%.	33 (37.5)	Confirmatory biopsy 18–24 months and every 2 years. PSA and DRE every 3 months in year 1, every 4 months in year 2, then every 6 months	PSA velocity, GS ≥4 + 3, PPC > 50%.	66	5.7 years	63.2% *	5-year rate of adverse histology and treatment-free probability, 22% and 70% *; 2 deaths for PC and 10 for any-cause	[65]
IR	Thostrup et al.	University of Copenhagen	107/451 (23.7%)	Prospective	≤T2, PSA <20 ng/mL, GS 4+3	39 (36.4)	PSA and DRE every 3 months for 2 years. TRUS-guided biopsy after 1 and 2 years. After 3 y PSA and DRE every 6 months.	Upgrade, upstage, PSADT	65.6 *	5.1 years	54.0% at 5 years	−	[66]
LR			180/451 (39.9%)								70.9% at 5 years		
IR	Thompson et al. **	St. Vincent’s Australia	59/650 (9.1%)	Retrospective	1-2 among: age < 55, >T2a, PSA > 10 ng/mL, low-volume GS 3+4, >20% of PPC	26 (44.1)	Confirmatory biopsy at 1 year, 1–2 years later, then every 3–5 years, PSA every 3 months for 3 years, then every 6 months; DRE every 6 months for 3 years, then annually	Upstage, upgrade in pattern 4, volume progression	63	55 months	60.3% *	CSS and OS, 100%	[67]
IR	Godtman et al.	Goteborg	104/474 (22.0%)	Retrospective	<71 y, ≤T2, PSA <20 ng/mL, GS 7	−	Confirmatory biopsy every 2–3 years, DRE and PSA every 3-6 months (up to 12 months).	Upgrade, upstage, PSA DT	66 *	8.0 years *	41% at 10-years, and 13% at 15-years	10-year and 15-year CSS, 98% and 90% for IR; 10-year and 15-year OS, 80% and 51% *; 10- and 15-year MFS, 99% and 93% *	[68]
LR			126/474 (27%)								42% at 10-years, and 27% at 15-years	10-y and 15-year CSS, 100% and 94%	
IR	Butler et al.	SEER	3223	Retrospective	≤T2, PSA <20 ng/mL, GS 7	−	−	−	67.9	39 months	−	5 y CSS, 98.9%; 5-year OS, 90.6%	[69]
IR	Thomsen et al.	2 Danish cohort	259/936 (27.7%)	Retrospective	≤T2, PSA < 20 ng/mL, GS 4+3	−	−	−	66 *	7.5 years	73.5% at 5 years and 69% at 10-years	10-year CSS, 99.5%; 5-year OS, 95.8%; 10-year OS, 83.9%	[70]
LR			436/936 (46.6%)								64.5% at 5 years and 55.7% at 10-years	10-year CSS, 99.3%; 5-year OS, 95.2%; 10-year OS, 87.9%	
IR	Loeb et al.	National Prostate Cancer Register of Sweden	328/1729 (18.9%)	Retrospective	<70 y, ≤T2, PSA <20 ng/mL, GS 7	116 (35.4)	Confirmatory biopsy after 18 months, then every 1–2 years, PSA and DRE every 3 months.	Upgrade, PSA DT	64	5 years	59% at 5-years	−	[71]
LR			757/1729 (44%)								67% at 5-years		
IR	Yamamoto et al.	University of Toronto	211/980 (21.5%)	Prospective	≤T2, PSA <20 ng/mL, GS 7	−	Confirmatory biopsy at 1 years, then every 3–4 years. PSA every 3 months for 2 years, then every 6 months.	Upgrade, upstage, PSA DT	−	6.4 years	−	MFS, 93.4%	[72]
LR			769/980 (78.4%)									MFS, 98%	
IR	Bul et al.	Rotterdam and Helsinki	128/509 (25.1%)	Prospective	PSA <20 ng/mL, GS 7, <3 cores	29 (22.6)	At doctors’ discretion	−	67.4	7.4 years	30.3% at 10-years	10-year CSS, 96.1%; 10-year OS, 64.5%; 10-year MFS, 96.4%	[73]
LR			381/509 (74.9%)						67.6		49.7% at 10-years	10-year CSS, 99.1%; 10-year OS, 79%; 10-year MFS, 99.7%	
IR	Herden et al.	University Hospital Cologne	82/482 (17.5%)	Prospective	≤T2, PSA <20 ng/mL, GS 7	30 (36.6)	Confirmatory biopsy at 1 year, then every 3 years. PSA every 3 months for 2 years, then every 6 months.	Upgrade, upstage, PSA DT	69.3	27.4 months	75.6%	CSS, OS, and MFS, 100%	[74]
LR			142/482 (30.3%)						68.2		78.9%	CSS, OS, and MFS, 100%	
IR	Shelton et al.	Multi-institutional	70/548 (12.7%)	Retrospective	<75 y, ≤T2, PSA <20 ng/mL, GS 7	33 (47.1)	At doctors’ discretion	−	−	3.35 years	59.1%	−	[75]
LR			218/548 (39.8%)								64.4%		
IR	Coperberg et al.	UCSF	90/466 (19.3%)	Prospective	≤T2, PSA < 20 ng/mL, GS < 8, CAPRA < 6	29 (32.2)	Biopsies every 12-24 months, DRE and PSA every 3 months, TRUS every 6–12 months.	−	65	51 months	61% at 4-years	−	[76]
LR			376/466 (80.7%)						62.3	47 months	54% at -4 years		
IR	Nyame et al.	Cleveland Clinic	108/635 (17.0%)	Retrospective	≤T2, PSA <20 ng/mL, GS 4+3	68 (63.0)	Confirmatory biopsy within 1 year, PSA and DRE every 6–12 months, and every 1–3 years.	Upgrade, upstage	68.6	44.2 months	94.8% at 5 years and 88.4% at 10 years for IR/HR	5-year and 10-year CSS 100%; 5- and 10-year OS, 95.6% and 77%; 5 and 10-year MFS 99.0% and 99% for IR/HR	[77]
LR			301/635 (47.4%)						65.1	51.2 months	97.7% at 5 years and 90.1% at 10 years for VLR/LR	5- and 10-year CSS 100%, 5- and 10-year OS 98.4% and 96.5%; 5- and 10-year MFS 99.2% and 97.4% for VLR/LR	
IR	Newcomb et al. **	Canary PASS	115/905 (13.0%)	Prospective	≤T2c, PSA 10–20 ng/mL, GS 7	56 (6.5)	Confirmatory biopsy after 1, 2, 4 and 6 years, PSA every 3 months, DRE at every 6 months.	Upgrade, volume, PPC	63	28 months	72.2%	CSS, 100%	[78]
IR	Carlsson et al. **	Memorial Sloan Kettering Center	219	Retrospective	GS 3+4	219 (100)	Confirmatory biopsy, PSA and DRE every 6 months.	Upgrade, upstage	67	3.1 years	61% at 5 years and 49% at 10 years	CSS and MFS, 100%	[79]

CSS, cancer-specific survival; DRE digital rectal examination; HR, high-risk; GS, Gleason score; IR, intermediate-risk; LR, low-risk; OS, overall survival; PPC, percentage positive biopsy cores; PSA, prostate-specific antigen; PSADT, PSA doubling time. * For all cohort. ** Do not differentiate between low and intermediate risk.

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
