# Peer review of "Active Surveillance in Intermediate-Risk Prostate Cancer: A Review of the Current Data"

_cancers, 2022, doi:10.3390/cancers14174161_

Round 1

Reviewer 1 Report

The study tried to summarize the current data on patients with intermediate-risk PC under AS, recent findings, and discusses future directions.

The logic of this paper is clear and the arguments are detailed, but there is still limitation.

Major issue:

The authors need to provide intuitive figures and tables in the paper to confirm that survival outcomes for intermediate-risk men on AS are comparable to low-risk in short-term and medium-term follow-up, but poorer in the long term.

Author Response

Major issue:

The authors need to provide intuitive figures and tables in the paper to confirm that survival outcomes for intermediate-risk men on AS are comparable to low-risk in short-term and medium-term follow-up, but poorer in the long term.

According to the comment, we have added this information to Table 2.

Reviewer 2 Report

1. Tables 1 and 2 mentioned in page 2 and page 7 cannot be found in the current manuscript.

2. The common definition of AS must be described at the beginning. Then, the authors should clarify the consistency or discrepancy of AS used in different studies in this review.

 3. In page 4, on the multivariable analysis, a PSA level of 10–20 ng/mL (vs PSA<10 ng/mL) presented lower odds of harboring an adverse pathology with an OR of 1.87 (95% CI 1.71–2.05, p<0.001), the lower odds was 1.87 ? Please clarify it.

 4. This review included various aspects of AS for patients with intermediate-risk prostate cancer, however, this is not a comprehensive and systematic review. The authors should provide some potential limitations for the current review.

Author Response

Reviewer: 2

  1. Tables 1 and 2 mentioned in page 2 and page 7 cannot be found in the current manuscript.

According to the comment, tables have been added.

  1. The common definition of AS must be described at the beginning. Then, the authors should clarify the consistency or discrepancy of AS used in different studies in this review.

According to the comment, this was added to the manuscript. (Introduction. Page 1).

  1. 3. In page 4, on the multivariable analysis, a PSA level of 10–20 ng/mL (vs PSA<10 ng/mL) presented lower odds of harboring an adverse pathology with an OR of 1.87 (95% CI 1.71–2.05, p<0.001), the lower odds was 1.87 ? Please clarify it.

According to the comment, the sentence was clarified in the manuscript. PSA 10-20 ng/mL (vs PSA< 10 ng/mL, OR 1.87) was lower than GS 2 (vs GS 1, OR 2.56).

  1. This review included various aspects of AS for patients with intermediate-risk prostate cancer, however, this is not a comprehensive and systematic review. The authors should provide some potential limitations for the current review.

According to the comment, this was added to the manuscript. (Discussion, Page 19).